# Universal Chern number statistics in random matrix fields

Or Swartzberg[1*], Michael Wilkinson[2,3†], and Omri Gat[1‡]

**1** Racah institute of Physics, Hebrew University, Jerusalem 91904, Israel
**2** Chan Zuckerberg Biohub, 499 Illinois Street, San Francisco, CA 94158, USA
**3** School of Mathematics and Statistics, The Open University, Walton Hall, Milton Keynes, MK7 6AA, England
*or.swartzberg@mail.huji.ac.il [†] m.wilkinson@open.ac.uk [‡] omrigat@mail.huji.ac.il

July 25, 2022

## Abstract

We investigate the probability distribution of Chern numbers (quantum Hall effect integers) for a parametric version of the GUE random matrix ensemble, which is a model for a chaotic or disordered system. The numerically-calculated single-band Chern number statistics agree well with predictions based on an earlier study [O. Gat and M. Wilkinson, *SciPost Phys.*, 10, 149, (2021)] of the statistics of the quantum adiabatic curvature, when the parametric correlation length is small. However, contrary to an earlier conjecture, we find that the gap Chern numbers are correlated, and that correlation is weak but slowly-decaying. Also, the statistics of weighted sums of Chern numbers for many bands differs markedly from predictions based upon the hypothesis that gap Chern numbers are uncorrelated. All our results are consistent with the universality hypothesis described in the earlier paper, including in the previously unstudied regime of large correlation length, where the Chern statistics is highly non-Gaussian.

# 1 Introduction

The discovery of the quantum Hall effect [1] was followed by an observation that the Hall conductance integer $N_n$ for systems with a band spectrum can be expressed as a Chern integer [2, 3]. This is a topological invariant that can be expressed as an integral of a curvature $\Omega_n$ over a closed surface $\mathcal{S}$ [4]:

$$N_n = \frac{1}{2\pi} \int_{\mathcal{S}} \mathrm{d}\boldsymbol{X} \; \Omega_n(\boldsymbol{X}) \; . \tag{1}$$

Here $n$ is an index labelling an isolated band of the spectrum. In the context of studies of the quantised Hall effect, the curvature is the quantum adiabatic curvature [5, 6] and the closed surface is a Brillouin zone [2], or a manifold specifying a boundary condition defined by complex phase factors [3].

It would be instructive to determine the properties of the integers $N_n$, and in particular their typical magnitude, when the number of bands is large. Moreover, because the Hall conductance integer is the sum of the values of $N_n$ for the filled bands, correlations between different values of $N_n$ are of interest: if the $N_n$ were uncorrelated random numbers, this might imply very large values of the Hall conductance. We might, therefore, anticipate that the there are correlations of the $N_n$ which reduce the magnitude of their sum.

Building on seminal work by Wigner [7] and Dyson [8], it is now understood that the spectra of complex quantum systems resemble those of samples from random matrix ensembles [9]. For our purposes, complex quantum systems are defined as those having many levels and no symmetries, localisation effects, or constants of motion. The Chern integers, however, are defined for families of Hamiltonians depending upon parameters— Bloch Hamiltonians depending on lattice momenta for the quantum Hall conductance. The random matrix theory approach has been extended to study correlations of energy levels and wave function of parameter-dependent complex systems [10–16].

A good starting point to understand the Chern numbers in complex systems is therefore to analyse the Chern numbers for parameter-dependent random matrix models. The goal of the present work is to examine the scaling, correlations, and universality of Chern numbers in parametric Gaussian Unitary Ensemble (GUE) models using Monte-Carlo calculations. Our work builds upon some earlier investigations which have applied parametric random matrix theory to the quantum adiabatic curvature and Chern integers. In particular, [17] addressed the statistics of Chern integers by calculating the statistics of degeneracies in a three-parameter parametric random matrix model: the results are consistent with the more refined investigation presented here. The single-point probability distribution of the curvature was obtained for a parametric random matrix model in [18]. The curvature in complex systems has also been studied using semiclassical approximations in [19, 20].

A recent paper, [21] by two of the present authors, referred to as paper I in the following, studied the statistics of the curvature, $\Omega_n(\boldsymbol{X})$, for parametric GUE models, and the results were used to propose expressions for the variance and the correlations of the Chern numbers. The comparison with the formulae proposed in paper I is informative: the prediction of the variance of the Chern integers proved to be very accurate, but there are significant deviations from the expression which was proposed for the correlations of weighted sums of Chern integers of different bands.

Namely, the prediction of paper I regarding multiband Chern number correlations was based on the hypothesis that the gap Chern numbers (that is, cumulative sums of band Chern numbers) are statistically independent. Our first main result is to show that this hypothesis is false: the correlations between the gap Chern numbers are small, but they decay slowly as a power law for increasing the spectral separation.

Our second main results concerns the band Chern number statistics when the parametric correlation length, equal to the inverse of the product of the density of states and the parametric sensitivity of the random matrix elements [21], is not small. When the parametric correlation length is taken to zero, the Chern number probability distribution is approximately Gaussian, and its variance tends to a constant limit when properly scaled. When the correlation length is not small, on the other hand, the distribution is far from Gaussian, and its moments depend in a complicated way on the correlation length. Nevertheless, we find this dependence becomes *universal* as the matrix size increases.

The paper is organised as follows. Section 2 defines the parametric GUE model to be studied. Section 3 reviews the results and hypotheses regarding the statistics of the Chern integers which were developed in paper I. Section 4 presents our results for the statistics of the Chern number of a single band. In section 5 we analyze the correlations of gap Chern numbers and their dependence on the spectral separation between the two gaps [21]. In section 6 we study the correlations of weighted sums of band Chern numbers, and their deviation from the predictions based on the results of paper I arising from gap Chern number correlation. Section 7 offers our conclusions from this work.

## 2 Parametric GUE models

To study the statistics of the Chern numbers, we used a model for fields of random matrices constructed using an approach considered in [11, 16]. We define an ensemble of fields of random matrices $H(\boldsymbol{p})$ over a parameter space $\mathcal{S}$. The ensemble is statistically homogeneous and isotropic in the parameter space, and it exhibits the following properties:

1. For any $\boldsymbol{p} \in \mathcal{S}$, the matrix $H(\boldsymbol{p})$ is a representative of the Gaussian Unitary Ensemble (GUE) of complex Hermitean random matrices, as defined in [8, 9]. The matrices have dimension $M$. The elements are independently Gaussian distributed, with zero mean, and their magnitude has unit variance.

2. The matrix element correlations are

$$\langle H_{ij}(\boldsymbol{p}_1)^* H_{ij}(\boldsymbol{p}_2) \rangle = c(|\boldsymbol{p}_1 - \boldsymbol{p}_2|) \tag{2}$$

   for some correlation function $c$ satisfying $c(0) = 1$ and positivity, and different matrix elements (other than those related by Hermiticty) are uncorrelated at different points in the parameter space.

In our work we took the manifold $\mathcal{S}$ to be the unit sphere embedded in three-space, and the distance between points $\boldsymbol{p}_1$, $\boldsymbol{p}_2$ on the sphere is taken as the Euclidean distance. Our numerical investigations included three different one-parameter families of correlation functions:

1. A Gaussian correlation function: $c(|\boldsymbol{p}_1 - \boldsymbol{p}_2|) = \exp(|\boldsymbol{p}_1 - \boldsymbol{p}_2|^2/(2r^2))$.

2. A Lorenzian correlation function: $c(|\boldsymbol{p}_1 - \boldsymbol{p}_2|) = 1/(1 + (|\boldsymbol{p}_1 - \boldsymbol{p}_2|/l)^2)$.

3. A *four-matrix model* of the form

$$H(\boldsymbol{r}) = \cos\alpha H_0 + \sin\alpha \, \boldsymbol{r} \cdot H_{\boldsymbol{r}} \tag{3}$$

   where $H_0$ is a GUE matrix, $H_{\boldsymbol{r}}$ is a vector of three GUE matrices and $\boldsymbol{r}$ is a unit vector on the sphere, implying that

$$c(|\boldsymbol{r}_1 - \boldsymbol{r}_2|) = \cos(\alpha)^2 + \sin(\alpha)^2 \frac{2 - |\boldsymbol{r}_1 - \boldsymbol{r}_2|^2}{2} \ . \tag{4}$$

According to the principles discussed in [10–12], the universal properties of this parametric random matrix model depend upon the density of states, $\rho$, and upon a tensor describing the sensitivity of the energy levels to displacements in the parameter space. For an isotropic model, this tensor is a represented by a diagonal matrix, and the parametric sensitivity is described by a single parameter, $\sigma^2$, which is the variance of first derivative of the levels:

$$\sigma^2 = \mathrm{var}\left(\frac{\partial E_n}{\partial p}\right) = -\frac{1}{2}\frac{\mathrm{d}^2 c}{\mathrm{d}p^2}\bigg|_{p=0} . \tag{5}$$

The parametric sensitivity can be any positive number in the Gaussian and Lorentzian families, and is between zero and one in the four-matrix model. In paper I there is a discussion of how a linear transformation of the parameter space of this model can be used to model systems in which the parameter space is not isotropic.

In our Monte-Carlo calculations, realisations of parametric GUE matrices on the sphere were randomly drawn, and for each realisation we numerically calculated the Chern number using an adaptive mesh version of the algorithm [22], dividing the sphere into triangles. This data was used to calculate the joint distribution of the Chern numbers for the three families of matrix-element correlations with different matrix sizes and correlation lengths.

## 3 Theoretical background

The correlation function of the curvature field $\Omega_n(\boldsymbol{p})$ was studied in [21]. It was argued that when the matrix size is large this correlation function may be expressed in the scaling form

$$\langle \Omega_n(p_1)\Omega_n(p_2)\rangle = (\rho\sigma)^4 f(\rho\sigma|\boldsymbol{p}_1 - \boldsymbol{p}_2|) \tag{6}$$

where $f(x)$ is a universal function, which approaches zero rapidly for large $|x|$. Equation (6) implies that the curvatures at points in parameter separated by a distance much larger than $1/(\rho\sigma)$ are uncorrelated. It follows, by taking the square of (1) and averaging, that when $\rho\sigma \gg 1$

$$\langle N_n^2\rangle = \frac{1}{2\pi}\mathcal{A}\sigma^2\rho^2\mathcal{I} \tag{7}$$

where $\mathcal{A}$ is the area of the surface $\mathcal{S}$ and $\mathcal{I} \approx 1.69$ is a universal constant obtained by integrating the function $f$. We note that $f(x)$ diverges as $1/x$ when $x$ tends to zero because the curvature distribution has infinite variance [18], but the integral of $f$ over the two-dimensional parameter space is finite.

Reference [21] also discussed statistics of a weighted average of $\Omega_n(\boldsymbol{p})$, defined by

$$\begin{aligned}\bar{\Omega}_\epsilon(E, \boldsymbol{p}) &= \sum_n \Omega_n(\boldsymbol{p})\delta_\epsilon(E - E_n)\\ \delta_\epsilon(E) &= \frac{1}{\sqrt{2\pi}\epsilon}\exp(-E^2/2\epsilon^2) .\end{aligned} \tag{8}$$

The correlation function

$$\mathcal{C}_\epsilon(\Delta E, \Delta\boldsymbol{p}) \equiv \langle \bar{\Omega}_\epsilon(E + \Delta E, \boldsymbol{p} + \Delta\boldsymbol{p})\bar{\Omega}_\epsilon(E, \boldsymbol{p})\rangle \tag{9}$$

was found in paper I to take the scaling form

$$\mathcal{C}_\epsilon(\Delta E, \Delta\boldsymbol{p}) = \frac{\pi^{3/2}}{6}\frac{\sigma^4\rho^3}{\epsilon^3}g(\rho\sigma|\Delta\boldsymbol{p}|, \Delta E/\epsilon) \tag{10}$$

in the limit $M \to \infty$, with $g$ a universal function of two variables, which was calculated exactly in the case where $\Delta p = 0$.

The fact that $\mathcal{C}_\epsilon(\Delta E, \Delta p)$ approaches zero rapidly as $\epsilon$ increases indicates a high degree of cancellation of the values of $\Omega_n(\boldsymbol{X})$ associated with different bands. In paper I it was argued that the fact that $\mathcal{C}_\epsilon \sim \epsilon^{-3}$ is consistent with the Chern integers having a two-point correlation of the form

$$\langle N_n N_m \rangle - \langle N_n \rangle \langle N_m \rangle = \frac{1}{2} \operatorname{var}(N_n) \left[ 2\delta_{nm} - \delta_{n,m+1} - \delta_{n,m-1} \right] . \tag{11}$$

We can describe this relation more succinctly with the help of the gap Chern numbers $G_n$, defined by

$$G_n = \sum_{i=1}^{n} N_i , \qquad 0 \le n \le N , \tag{12}$$

so that $N_n = G_n - G_{n-1}$, and therefore (11) is consistent with the hypothesis that the gap Chern numbers are uncorrelated:

$$\langle G_n G_m \rangle = 2 \operatorname{var}(N_n) \delta_{nm} . \tag{13}$$

In paper I this hypothesis was applied to the statistics of the weighted sum of Chern numbers in an energy window of width $\epsilon$,

$$\bar{N}_\epsilon(E) = \sum_n N_n \delta_\epsilon(E - \langle E_n \rangle) . \tag{14}$$

where $\langle E_n \rangle$ is an average of $E_n(\boldsymbol{p})$ over the parameter space. We choose $\epsilon$ much larger than the mean level spacing, but small enough that the density of states does not change appreciably inside the energy window. A direct estimate of the variance of (14) involves integrating the correlation function (10) over $\Delta p$, and leads to the prediction that $\langle \bar{N}_\epsilon^2 \rangle \sim \epsilon^{-3}$. However, the function $g(\Delta X, \Delta E)$ was not sufficiently well characterised in paper I to allow this integral to be estimated reliably. An alternative approach estimates $\langle \bar{N}_\epsilon^2 \rangle$ using (7) and the independent gap Chern number hypothesis, (11) yielding,

$$\langle \bar{N}_\epsilon^2 \rangle \sim \frac{\langle N^2 \rangle}{2\pi \epsilon^2} F(X) \tag{15}$$

where $X = 1/(2\epsilon^2 \rho^2)$, and

$$F(X) = \sum_{n=-\infty}^{\infty} \exp(-Xn^2)[\exp(-Xn^2) - \exp(-X(n+1)^2)/2 - \exp(-X(n-1)^2)/2] . \tag{16}$$

As explained in paper I, the expressions (15), (16) can be further simplified when $\epsilon \rho \gg 1$ to obtain

$$\langle \bar{N}_\epsilon^2 \rangle \approx \frac{3\mathcal{I}}{128\sqrt{\pi}} \frac{\mathcal{A}\rho\sigma^2}{\epsilon^3} , \tag{17}$$

exhibiting the the same $\epsilon^{-3}$ as in (10). Our numerical results are consistent with the prediction $\langle \bar{N}_\epsilon^2 \rangle \sim \epsilon^{-3}$, over a small range of $\epsilon$. However, it is found that the coefficient multiplying $\epsilon^{-3}$ in (17) is not correct.

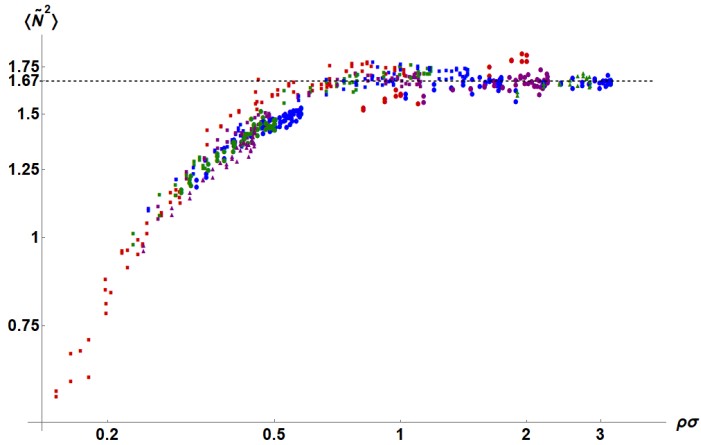

Figure 1: Double logarithmic plot of the variance of Chern numbers of the bands in the bulk of the spectrum of random matrix field models, with different sizes from $4 \times 4$ to $50 \times 50$, and matrix element correlation functions, calculated in Monte-Carlo simulation as a function of the inverse parametric correlation length $\rho\sigma$. The scaled Chern number variance $\langle \tilde{N}_n^2 \rangle \equiv 2\pi \langle N_n^2 \rangle / \rho^2 \sigma^2 \mathcal{A}$ approaches the universal constant $\mathcal{I} \approx 1.67$ when $\rho\sigma$ is large. Symbol shapes correspond to Gaussian (circles), Lorentzian (triangles) and four-matrix (square) matrix-element correlations, and colours to matrix sizes dimensions. The data collapse shows that band Chern number variance is a universal function of $\rho\sigma$, even when this variable takes small values, due to the scale factor ($r$ or $l$ in the Gaussian and Lorentzian random-matrix models respectively) being large.

## 4  Band Chern number statistics

We calculated the variance of Chern number for a large number of realisations of the random-matrix fields on the two-sphere, with dimensions between $M = 4$ and $M = 50$, with elements correlated according to the three families listed in section 2, and with several choices of scale factors, $r$ or $l$. Figure 1 shows the scaled variance of the Chern integers,

$$\langle \tilde{N}_n^2 \rangle = \frac{2\pi \langle N_n^2 \rangle}{\rho^2 \sigma^2 \mathcal{A}} \ , \tag{18}$$

plotted as a function of $\rho\sigma$. As expected, when $\rho\sigma \gg 1$, $\langle \tilde{N}_n^2 \rangle$ approaches an asymptotic universal constant value $\mathcal{I} \approx 1.67$, irrespective of the shape of the correlation function, consistent with (7) and the estimate of $\mathcal{I} \approx 1.69$ based on the results of [21]. We note however that the asymptotic regime of 'large' $\rho\sigma$—small correlation length—starts already in when $\rho\sigma \gtrsim 0.7$, so that the estimate that the parametric correlation length $\sim 1/(\rho\sigma)$ involves a small proportionality factor. This conclusion is also consistent with the finding of [21] that $xf(x)$, where $f(x)$ is the scaling function of (6) drops to half its maximal value already for $x \approx 0.15$.

In the regime of large $\rho\sigma$ we furthermore expect that the Chern-number distribution is Gaussian because when the parametric correlation length is small, the integral in (1) can be viewed as a sum of many nearly independent random variables, obtained by dividing the surface $\mathcal{S}$ into patches whose size is large compared to the parametric correlation length, $1/\rho\sigma$, but small compared to the entire surface.

We studied the convergence of the Chern-number distributions to Gaussian by calcu-

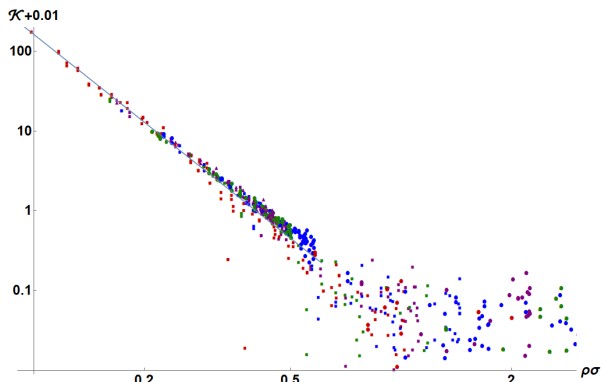

Figure 2: Double logarithmic plot of the (excess) kurtosis $\kappa = \langle N_n^4 \rangle / \langle N_n^2 \rangle^2 - 3$ of the Chern number distributions whose variance is shown in figure 1, as a function of $\rho\sigma$. The statistical uncertainty in the data is about 0.1, so that $\kappa$ values for $\rho\sigma \gtrsim 0.7$ are consistent with zero, consistent with a Gaussian Chern number distribution in this regime. Note that a constant shift of 0.01 has been added to all values of $\kappa$ to improve the clarity of the graph. The positive $\kappa$ values obtained for $\rho\sigma \lesssim 0.7$ show that the Chern number distribution is non-Gaussian in this regime, while the data collapse further supports the hypothesis that the distribution is universal for all $\rho\sigma$. The straight line is a power law fit $\kappa \sim (\ell_\kappa \rho\sigma)^{-\alpha}$, $\alpha = 3.66 \pm 0.05$, $\ell_\kappa = 2.49 \pm 0.05$ for $\rho\sigma \leq 0.7$.

lating the (excess) kurtosis

$$\kappa = \frac{\langle N_n^4 \rangle}{\langle N_n^2 \rangle^2} - 3 \qquad (19)$$

of the randomly generated Chern-number populations; we use the definition which makes $\kappa = 0$ in Gaussian distributions.

Our Monte-Carlo calculations of the kurtosis are shown on a double-logarithmic plot in figure 2. As expected, $\kappa$ becomes small as $\rho\sigma$ increases toward the small correlation length regime $\rho\sigma \gtrsim 0.7$. We note that the statistical uncertainty of our Monte-Carlo results is about $\pm 0.1$, so that values of $\kappa$ smaller than 0.1 are not statistically distinguishable from zero. On the other hand, very small values, and negative values of $\kappa$ can be randomly obtained, which cannot be conveniently plotted on a log-log graph. For this reason, figure 2 actually shows $\kappa + 0.1$, allowing it to keep most of the data points inside the viewing range. Thus, for $\rho\sigma \gtrsim 0.7$, our numerical results agree with the prediction that the Chern number distribution is Gaussian, and its variance is given by (7).

As expected these predictions are not valid when $\rho\sigma \lesssim 0.7$: Figures 1 and 2 show that in this regime the scaled Chern variance $\langle \tilde{N}_n^2 \rangle$ is an increasing function of $\rho\sigma$, and $\kappa$ is positive. Nevertheless, the data collapse seen in these figures is consistent with extended *universality*, where $\langle \tilde{N}_n^2 \rangle$, $\kappa$, and plausibly the entire Chern number distribution depends on $\rho\sigma$, but not on any other detail of the random matrix element distribution. We conjecture that the universal distribution is an exact asymptotic for matrix size $M \to \infty$, but the numerical calculation indicate that it is a good approximation already for moderate $M$.

As a last observation on single-band Chern number statistics, we note that while the dependence of the scaled Chern number variance on $\rho\sigma$ has no apparent structure for $\rho\sigma \lesssim 0.7$, the kurtosis follows the power law

$$\kappa \sim (\ell_\kappa \rho\sigma)^{-\alpha} , \qquad (20)$$

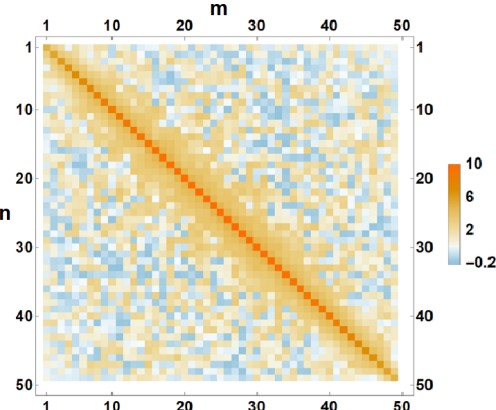

Figure 3: Plot of the covariance matrix $\langle G_n G_m \rangle$ of the gap Chern numbers of $50 \times 50$ matrix fields with a Gaussian matrix-element correlation function with correlation length $r = 1$, with values coloured according to the legend on the right. It is evident that the diagonal elements of the covariance matrix are much larger than the off-diagonal, but the main diagonal is surrounded by a thick diagonal band of positive correlations, beyond which the measured correlation are comparable with the statistical uncertainty, and are therefore consistent with zero.

where $\alpha = 3.43 \pm 0.05$, $\ell_\kappa = 2.39 \pm 0.05$ are universal constants. At the moment, we have no explanation of this result.

## 5 Gap Chern number correlations

In paper I it was argued that (11) is a plausible expression for correlations between Chern integers. It was shown that (11) is compatible with the $\epsilon^{-3}$ scaling of (10), and this expression appears to be the simplest hypothesis which is compatible with (10). However, we find that the numerical calculated correlation of the curvature of neighbouring bands has a small but statistically significant violation of (11). The assumption (11) is equivalent to the hypothesis that the gap Chern numbers are statistically independent.

Here we further test this hypothesis by calculating the gap Chern number correlations directly. Figure 3 shows a colormap of the covariance $\langle G_n G_m \rangle$ in a population of $50 \times 50$ matrix fields. Evidently, the covariance is positive in a thick diagonal band, even though the off-diagonal terms are much smaller than the diagonal terms of the covariance matrix.

The gap Chern number correlations was studied systematically using the Pearson correlation

$$g_{n,k} = \frac{\langle G_n G_{n-k} \rangle}{\sqrt{\langle G_n^2 \rangle \langle G_{n-k}^2 \rangle}} \ . \tag{21}$$

Our numerical results indicate that the correlation coefficients are independent of $n$ and have a universal power law dependence

$$g_{n,k} \sim \frac{g_1}{k^\gamma} \ , \tag{22}$$

with $g_1 = 0.18 \pm 0.01$, $\gamma = 0.62 \pm 0.01$, for all gap Chern number in the short-correlations regime $\rho\sigma \gtrsim 0.7$ (see Figure 4).

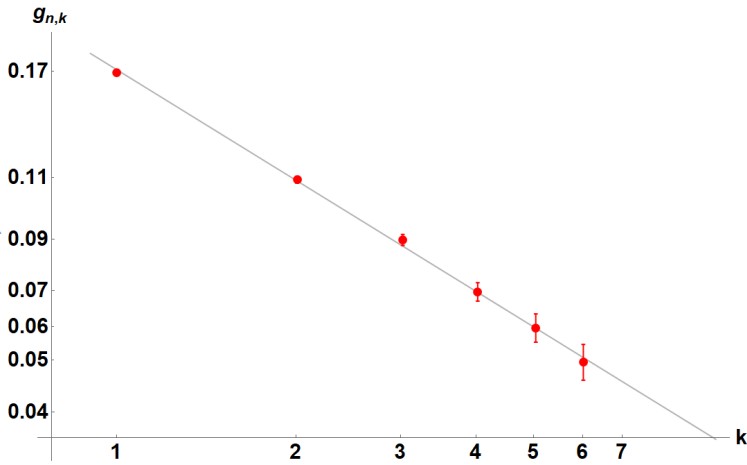

Figure 4: The off-diagonal correlation coefficient $g_{n,k}$, calculated from the gap Chern number distributions of all random matrix sizes and matrix-element correlations in the short correlations regime $\rho\sigma \geq 0.7$, and all base levels $n$, shown as a function of the offset $k$. The centres and error bars of each point show the mean and standard deviation (respectively) of the population of $g_{n,k}$ samples obtained in this way; the smallness of the standard deviations supports the hypothesis that the $g_{n,k}$ are universal coefficients independent of $n$. The universal function is well described by the power law fit $g_{n,k} \sim \frac{g_1}{k^\gamma}$ with $g_1 = 0.18 \pm 0.01$, $\gamma = 0.62 \pm 0.01$.

## 6  Weighted Chern number variance

We computed the variance of the weighted Chern number statistic, as defined in equation (14). We considered a range of values of $\epsilon$, taking $E = 0$ and using the largest matrices (dimension $M = 50$), in both the Gaussian and four-matrix models. Figure 5 shows the dependence of $\langle \bar{N}_\epsilon^2 \rangle$ on $\epsilon$, using a double-log scale for the Gaussian and four-matrix models, compared with values for these variances predicted by (15) and (16), and with the asymptotic formula (17), on the basis of the independent gap Chern number hypothesis (13).

Since the density of states is large in our calculations, the values based on (15) and (16) are well-approximated by (17). However, these predictions, based on the independent gap Chern number hypothesis, are not in agreement with the Monte-Carlo results. While (17) gives the correct order of magnitude of the weighted Chern number variance, the numerically evaluated results are greater by a factor as large as three: figure 6 shows the ratio of the numerically obtained $\langle \bar{N}_\epsilon^2 \rangle$ divided by the prediction based on (15), (16), for both the Gaussian and the four-matrix models. While we do not have a theoretical explanation for the behaviour of the ratio, the close agreement between the two models supports the universality hypothesis for the weighted Chern statistic as well.

It is interesting that while the gap Chern number correlations studied in section 5 are small, the deviations from (15) and (16) are quite significant. This may be an indication that there are non-trivial multi-level correlations between Chern numbers, but further study of this question is beyond the present scope.

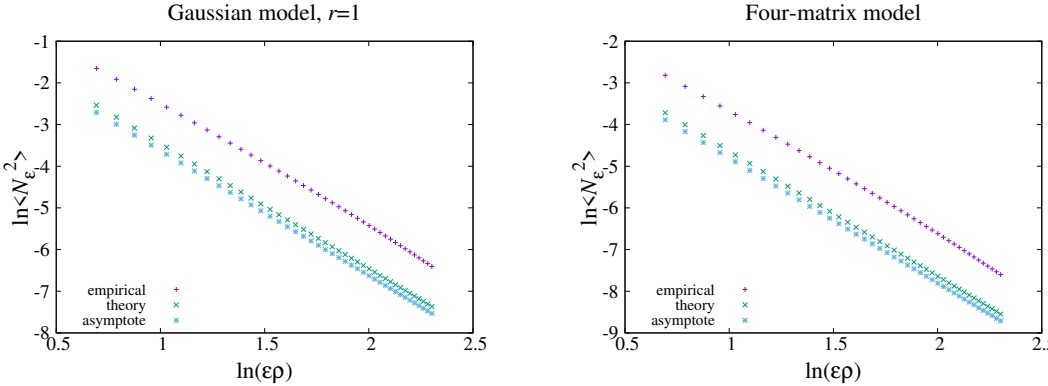

Figure 5:   Double logarithmic plot of the weight Chern number variance $\langle \bar{N}_\epsilon^2 \rangle$ versus the product of the density of states$\rho$ and the with $\epsilon$ of the energy window. The purple + symbols show $\langle \bar{N}_\epsilon^2 \rangle$ calculated from Monte-Carlo simulation of the random matrix model, with energy window centred at $E = 0$. The green $\times$ and blue $*$ symbols show the values of $\langle \bar{N}_\epsilon^2 \rangle$ expected on the basis of independent gap Chern number hypothesis, formula (15), and its asymptotic approximation (17), respectively. When $\epsilon\rho$ is large, (15) agrees with (17) as expected, but both are significantly smaller than the numerical results.

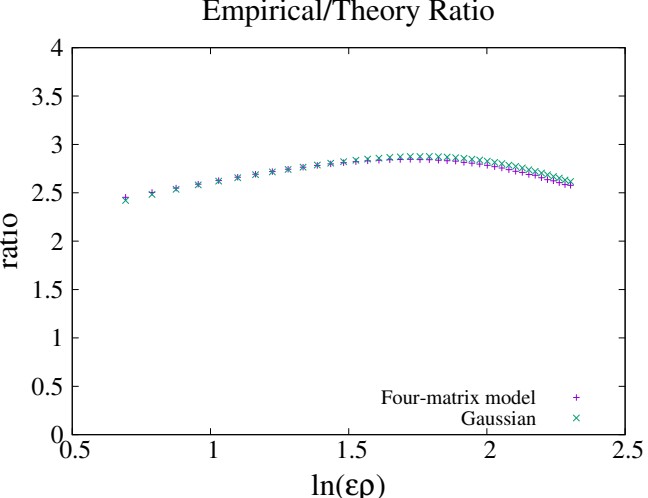

Figure 6:   Values of $\langle \bar{N}_\epsilon^2 \rangle$ numerically calculated for the four-matrix model (purple + symbols) and Gaussian correlations model (green $\times$) divided by the prediction based on formula (15) and (16), shown as a function of $\ln(\rho\epsilon)$. The significant deviation of this quotient from one is a consequence of the failure of the independent gap Chern number hypothesis, which spoils the strong cancellation in (15) and (16). The close agreement between the ratios obtained for the two random matrix models is consistent with the Chern-number-statistics universality hypothesis.

# 7 Conclusions

The present numerical study of the Chern numbers of parametric random matrix models examined three types of statistics: moments of the single-band Chern number distribution, correlations of the gap Chern numbers, and the variance of weighted sums of band Chern numbers.

The single-band statistics confirmed that when the parametric correlation length is small, the band Chern number distribution approaches a Gaussian, with variance inversely proportional to the square of the correlation length, and that the proportionality factor is universal, consistent with the results of paper I [21]. It showed furthermore that the distribution is non-Gaussian but universal in the previously unexplored long correlations regime, and that the kurtosis of the distribution depends as a power law on the correlation length in this regime.

We found that the gap Chern number are correlated, refuting an earlier hypothesis. The gap Chern number correlation are weak, but decay slowly as a power law when the spectral separation between the gaps increases.

When we looked at the variance of a weighted sum of Chern numbers, however, we found that there was a very significant difference from the predictions based upon assuming independent gap Chern numbers. One of the motivations for looking at statistics of Chern numbers was the hypothesis that there are cancellation effects, which reduce the variance of a sum of Chern numbers. We found that the variance of the weighted sum was larger than the prediction in paper I, indicating that the cancellation effects are weaker than anticipated.

We were motivated to perform this study by the success of random matrix models as exemplars of universal properties of complex quantum systems. Our results do show how random matrix models for Chern integers can be successfully quantified within the framework of a 'universality' hypothesis. However, it is not clear to what extent these random matrix models are representative of the quantised Hall effect in physically realistic models. In particular, when the dimension of the random matrix is large, then sum of the band Chern numbers may be a very large number. It is desirable to complement this present work with studies of Chern numbers in physically realisable models for complex quantum systems.

## Acknowledgements

MW is grateful for the generous support of the Racah Institute, who funded a visit to Israel.

**Funding information** This work was supported by the German-Israeli Foundation for financial support under grant number GIF I-1499-303.7/2019.

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
