# Peer review of "Universal Chern number statistics in random matrix fields"

_SciPost Physics_

## Round 1 · Referee Report · Anonymous (Referee 1) · 2022-8-21

Strengths

1) The paper is well written, with detailed abstract, introduction, and conclusion of the results presented.

2) Throurough study of the band Chern numbers as a function of the parametric correlation length.

3) The authors show that the gap Chern numbers in parametric GUE models are correlated.

Weaknesses

1) The method for the calculation of the Chern numbers is discussed too briefly.

2) No code is available to reproduce their results.

3) Some of the results presented are not satisfactorily explained, and there is no connection between them and implications for physical systems.

Report

#About my expertise: I am an expert in the topological aspects of condensed matter systems. I only have a general understanding of RMT.

In this paper, the authors study the (band and gap) Chern number probability distribution in random parametric Gaussian Unitary Ensemble (GUE) models, considering three different functions for the correlations between the matrix elements. Understanding the correlations could shed light on the possible values for the Hall conductance as uncorrelated band Chern numbers could imply large values for it.

The authors compute the Chern numbers via a Monte Carlo approach and study their properties as a function of matrix size and the model parameter rho*sigma‑‑the product of the density of states and the parametric sensitivity. In particular, they show results for the variance of band Chern numbers, the (excess) kurtosis of the randomly generated Chern-number populations, the covariance matrix of the gap Chern numbers, and weight Chern number variance.

They found numerically that the variance of the (scaled) band Chern numbers tend to a universal constant value (~1.67) for rho*sigma > 1 (corresponding to small correlation lengths.) This is in agreement with the results presented in the aprevioud paper by some of the same authors. The kurtosis shows consistent results. The band Chern number statistics
show a universal dependence on the parametric correlation length for larger matrices. The gap Chern numbers are correlated, albeit weakly, with Pearson correlations with a universal power law dependence. The authors emphasize that this is in contrast with an assumption made by the authors in their previous paper (Ref. [21]).

This paper was an interesting read for this referee, with numerical insights into the Chern numbers of parametric GUE models. I would recommend the publication of this paper if the questions in the requested changes sections are addressed.

Requested changes

1) Some of the results presented lack a deeper physical or mathematical understanding. For example, the scaled Chern number variance approaches a universal constant (1.67) for small correlation lengths. Is there an explanation connected with quantum systems?

2) The kurtosis follows the power law with universal constants, but the authors have "no explanation of this result." I think it would be Ok to offer speculations about the origin of these results.

3) The weighted Chern number variance shows deviation from previous results by some of the authors. However, no further insight is provided. Could the authors expand on this issue?

4) The Pearson correlation for the gap Chern number n follows a universal function independent of n. The authors could offer further insight into the origin of this universality.

5) The method for the calculation of the Chern numbers is discussed very briefly. I would appreciate a more detailed explanation and will strongly encourage the authors to make their code openly available. Two of the acceptance criteria are "Provide (directly in appendices, or via links to external repositories) all reproducibility-enabling resources," and "Provide sufficient details (inside the bulk sections or in appendices) so that arguments and derivations."

6) In the introduction, the authors first motivate the study based on the effect of correlated Chern numbers on the value of the Hall conductance in physical systems. In the conclusions, however, the authors stated that "it is not clear to what extent these random matrix models are representative of the quantized Hall effect in physically realistic models." I would appreciate an expanded discussion about this. One of the acceptance criteria is that the paper should "Contain a clear conclusion summarizing the results (with objective statements on their reach and limitations) and offering perspectives for future work." The authors could discuss potential physically realistic relevant models where these questions can be explored. Additionally, do the authors' ambition to extend these types of studies to topological invariants beyond Chern numbers?

  • validity: high
  • significance: ok
  • originality: ok
  • clarity: good
  • formatting: good
  • grammar: excellent

Author:  Michael Wilkinson  on 2023-03-29  [id 3519]

(in reply to Report 1 on 2022-08-21)

We thank the referee for their careful appraisal of our work.

The referee asks for more details of the numerical methods, and for the codes to be made publicly available. We have added an appendix, which explains the numerical approaches in some detail, and we have included a URL where the codes are available:

https://github.com/orswartzberg/Numerical-calculation-of-Chern-numbers

The referee also asks for more detail on the physical context. We have revised the Introduction accordingly. Our revision includes more general references about recent developments in topological approaches to condensed matter physics. We have also expanded upon the physical motivation for our investigation (in the Introduction) and upon the remaining unresolved issues (Conclusion).

Regarding the specific requested changes:

  1. The universal variance coefficient ~1.67 is expected to be applicable to complex quantum systems under a universality hypothesis, as discussed in the introduction. It is related to an integral of a correlation function in ref. [21], but we were not able to make an analytic estimate.

  2. Our attempts to explain the power-law relation for kurtosis were unsuccessful. Explaining this dead-end would be quite involved. We think there is nothing that we can usefully add to what is written.

  3. The text in section 6 states that the discrepancy between our numerical results and the prediction in ref. [21] is a result of correlations between the gap Chern numbers, whicg were assumed to be absent in the earlier work.

  4. The independence upon n is a consequence of the fact that random matrix models are ‘democratic’ in their treatment of individual states, and complex quantum systems should share this property. We have added a comment (below (21)) to that effect.

  5. As discussed above, we have added an appendix explaining the methods, and made the code available.

  6. We have expanded upon the un-resolved issues concerning the use of Chern numbers to describe the Hall effect in complex systems in the final paragraph of the Conclusions, indicating that we retain an ambition to address these in subsequent work.

---

## Round 1 · Referee Report · Anonymous (Referee 2) · 2023-1-28

Strengths

Presents very interesting numerical results on correlations between Chern numbers of bands in multi band systems

Weaknesses

Not enough detail on the numerical procedure. No analytic understanding.

Report

OK to publish pending requested changes below.

Requested changes

Given that this is a mainly numerical work, authors should give more details on how the numerics was done.

Some speculation on what might be going on would be welcome (but is optional)

Also optional: consider applying this to the mini bands in Moire systems (e.g. twisted bilayer graphene). That could be a fruitful setting in which to apply these results.

---

## Round 1 · Referee Report · Anonymous (Referee 2) · 2023-1-28

Strengths

Presents very interesting numerical results on correlations between Chern numbers of different bands in multiband systems.

Weaknesses

There isn't much in the way of conceptual understanding, and not enough detail on the actual numerical procedure used.

Report

Yes, subject to requested changes below

Requested changes

(1) Since the results are mainly numerical, it would be a good idea to give more details on the numerical procedure.
(2) Optional: speculations as to what might be giving rise to the observed correlations?
(3) Optional: consider applying this to the minibands in Moire systems? e.g. twisted bilayer graphene? That would be a very topical setting in which this analysis could be fruitfully applied.

---

## Round 1 · Referee Report · Anonymous (Referee 3) · 2023-1-29

Strengths

  1. The paper is well-written and the results are clearly derived. Notation is appropriate and not confusing. The figures are helpful in representing the computations described in the text and are clearly constructed and presented.

  2. This paper represents an extension of previous work pioneered by one of the coauthors (MW) on Chern numbers for parameterized random ensembles of unitary matrices and includes new results, i.e. the correlation of the Chern numbers.

  3. There are interesting findings of universality in the Chern number statistics.

Weaknesses

  1. Despite being motivated by the quantum Hall effect, there is little effort to make any connection to the physics of the QHE other than the relevance of Chern numbers.

  2. There is inadequate discussion of the numerical methods used in obtaining the results presented.

Report

This is an well-motivated investigation which presents new results in the study of parameterized random matrix ensembles, a field with one of the coauthors (MW) pioneered.

As a numerical investigation of a clearly defined problem in mathematics, this paper is compelling. From the physics perspective, in order to better connect with the quantum Hall effect, which seems to be an objective of the authors, some discussion of how the parameterized RM ensembles in section 2 might be relevant to actual quantum Hall systems such as ballistic 2DEG systems like GaAs, or graphene (or possibly Floquet Chern systems) would be appropriate. However, the manuscript stands well on its own as it is, and no changes in this regard are required.

The most unsatisfying aspect is in the inadequate description of the numerical methods used in obtaining the results. A brief appendix describing these methods would be most welcome.

Requested changes

The only change I would request is a fuller description of the numerical methods used. This would be helpful to readers who might wish to follow up on this work. A brief appendix describing the numerics would be much appreciated.

  • validity: high
  • significance: good
  • originality: high
  • clarity: high
  • formatting: perfect
  • grammar: perfect

Author:  Michael Wilkinson  on 2023-03-29  [id 3518]

(in reply to Report 4 on 2023-01-29)

We thanks the referee for their generous comments and helpful criticisms.

We have expanded the Introduction to strengthen the physical motivation for the paper, as discussed in the response to referee 1.

We have also included an appendix giving a full description of the novel aspects of the numerical approach, and we have made the numerical codes available: see

https://github.com/orswartzberg/Numerical-calculation-of-Chern-numbers

---

## Editorial Decision

resubmitted